# Liquid Biopsy-Based Biosensors for MRD Detection and Treatment Monitoring in Non-Small Cell Lung Cancer (NSCLC)

**DOI:** 10.3390/bios11100394

**Published:** 2021-10-15

**Authors:** Parvaneh Sardarabadi, Amir Asri Kojabad, Davod Jafari, Cheng-Hsien Liu

**Affiliations:** 1Institute of Nanoengineering and Microsystems, National Tsing Hua University, Hsinchu 30044, Taiwan; s107035860@m107.nthu.edu.tw; 2Department of Hematology, School of Allied Medical Sciences, Iran University of Medical Sciences, Tehran 14535, Iran; asrikojabad.a@iums.ac.ir; 3Department of Medical Biotechnology, School of Allied Medicine, Iran University of Medical Sciences, Tehran 14535, Iran; jafari.d@tak.iums.com; 4Department of Power Mechanical Engineering, National Tsing Hua University, Hsinchu 30044, Taiwan

**Keywords:** bio-sensor, minimal residual disease, liquid biopsy, non-small cell lung cancer

## Abstract

Globally, non-small cell lung cancer (NSCLC) is the leading cause of cancer deaths. Despite advancements in chemotherapy and targeted therapies, the 5-year survival rate has remained at 16% for the past forty years. Minimal residual disease (MRD) is described as the existence of either isolated tumour cells or circulating tumour cells in biological liquid of patients after removal of the primary tumour without any clinical signs of cancer. Recently, liquid biopsy has been promising as a non-invasive method of disease monitoring and treatment guidelines as an MRD marker. Liquid biopsy could be used to detect and assess earlier stages of NSCLC, post-treatment MRD, resistance to targeted therapies, immune checkpoint inhibitors (ICIs) and tumour mutational burden. MRD surveillance has been proposed as a potential marker for lung cancer relapse. Principally, biosensors provide the quantitative analysis of various materials by converting biological functions into quantifiable signals. Biosensors are usually operated to detect antibodies, enzymes, DNA, RNA, extracellular vesicles (EVs) and whole cells. Here, we present a category of biosensors based on the signal transduction method for identifying biosensor-based biomarkers in liquid biopsy specimens to monitor lung cancer treatment.

## 1. Introduction

Today lung cancer is one of the deadliest forms of cancer, with an extremely high mortality rate of around 18% of all cancer-related deaths [1]. According to the histological classification, the two main lung cancer types are non-small cell lung cancer (NSCLC) and small cell lung cancer (SCLC). NSCLC is the dominant type of lung cancer with 40% more propagation than other lung cancers. Therefore, the prognosis of lung cancer in the early stages is considered a good chance to prevent cancer progression. According to the TNM stage groupings, four stages determine NSCLC disease: Stage 0 (hidden), in this stage cancer cells are found in sputum or bronchial washings, but imaging techniques or bronchoscopy cannot identify them clearly; also cancer may have extended to other parts of the body; Stage I (formation of cancer), cancer has not spread to the lymph nodes yet; Stage II, cancer has spread to the main bronchus; Stage III, cancer has spread to some parts of the body, but without any signs of metastasis; Stage IV, cancer has spread to multiple places either in one or more organs. Around 80% of cases that are positive for NSCLC have beforehand progressed to advanced stages.

Based on the National Comprehensive Cancer Network (NCCN) guidelines for high-risk early stage (stage I to III) NSCLC, surgery is the primary treatment for NSCLC [2,3,4]. Up to 60% of stage II-III patients who receive surgery will eventually develop recurrence. In addition to surgery, other treatments such as localised radiotherapy and chemotherapy are used for localised disease. MRD, which occurs when small amounts of cancer cells persist in the tissue or body, is critical during treatment. This MRD gradually spreads and causes recurrences that have poor outcomes. Adjuvant therapy is used as a standard treatment for non-metastatic cancer, stage II-IIIA NSCLC. Adjuvant therapy can include chemotherapy, radiation therapy, and targeted therapy. According to clinical studies, for patients with stage II, the post-surgery 5-year disease-free survival (DFS) rates are under 50% [5]. This percentage is 46% for grade IIA NSCLC and 36% for grade IIB [4].

Regarding the published data, the detection and assessment of earlier stages of tumours prolong the overall survival (OS) of resected NSCLC patients [6,7]. However, disease relapse is thought to be caused by treatment-resistant cells that are not eliminated after completing a course of treatment [8]. These remaining cells, known as minimal/measurable residual diseases (MRD), are undetectable by standard detection methods. MRD refers to the cells that persist after therapy and are resistant to conventional therapy. Several studies have proven that MRD can be a prognostic factor for relapse in these patients [9]. MRD is well established in malignancies of the haematological and lymphoid organs. However, additional research is needed in solid tissue cancers such as lung cancer before being used in clinical practice.

The biopsy is a standard method for accessing the tumour tissue, an invasive and unpleasant procedure for the patient. Furthermore, there is variability in the outcomes of separate biopsies [10]. Biopsies are more challenging for high-risk patients, individuals with early lung cancer, patients with advanced-stage disease and other conditions in which tumour tissues cannot be obtained [11,12]. With considering all side effects of tissue biopsy, a novel approach (liquid biopsy) has been established. Liquid biopsy is defined as sampling and analysing biomarkers such as exosomes, circulating tumour cells (CTCs) and circulating free DNA (cfDNA) in primary blood or other biological fluids. The abilities such as continuous monitoring, convenient, non-invasive and applicable for early prognosis have made it a leading-edge tool for the early prognosis of NSCLC. Monitoring circulating tumour DNA (ctDNA), CTCs and EVs in fluid secretion gives more opportunities to announce recurrence possibilities and reactions to treatment in lung cancer patients. Moreover, heterogeneity and mutation in NSCLC are detectable through ctDNA extracted from body fluids [13]. MRD quantification and monitoring in solid tumours have been limited to liquid biopsy. Unlike haematological neoplasms, access to tissue is not always available in solid tumours. Routine MRD measuring methods such as next-generation sequencing (NGS) and PCR-based tests are costly as well trained technicians and utilizing sophisticated instruments are required [14,15,16]. Morphological techniques are less sensitive for detecting low levels of MRD in biopsy fluid and do not fulfil clinical requirements. Table 1 shows the advantages and disadvantages of routine methods.

Recently, targeted therapies and immune checkpoint inhibitors (ICIs) have demonstrated remarkable results for the consolidation treatment of NSCLC [17,18]. Some immunotherapies drugs have been FDA approved to detect programmed death-ligand 1 (PD-L1) expression level and block PD-1 ligands functions on cancer cells [19]. As a result, the presence or absence of PD-L1 in the blood of patients with advanced-stage lung cancer might influence the course of treatment or its continuation. People who have MRD can also benefit from checkpoint inhibitors. It was initially used in patients with advanced melanoma in the 1990s, and later it was expanded to other solid malignancies [20].

Identifying the appropriate biomarkers is an approach to improve the efficiency and reliability of employing MRD in clinical practices. MRD biomarkers can be used for early diagnosis of NSCLC as well as monitor response to treatment. Nowadays, in progressive cancer disease, the detection of genomic alteration for targeted therapy has been deemed the standard of care to improve patient survival. However, biomarkers for early diagnosis can be different from biomarkers for monitoring or MRD. Furthermore, the type of biomarker is unknown in the early diagnosis; while we are looking for a known biomarker during biomarker monitoring, not every biomarker may be suitable for MRD purposes [21,22,23].

The development of sensitive, simple, low-cost and accessible patient bedside procedures (POCTs) can help to accelerate the clinical application of liquid biopsy in MRD. Alternatively, ultra-sensitive biosensor-based methods are introduced to detect residual diseases in the postoperative setting. As a powerful tool in biology and biochemistry, biosensors have an appreciable potential to meet some of this field’s needs. A biosensor is a device designed to detect biological substances. Moreover, it can identify chemical compounds by employing specific biological processes mediated by individual enzymes, immunosystems, tissues, organelles or entire cells, mostly via electrical, thermal or optical signals [24]. Each biosensor is made up of three parts: First, the bio receptor for the analyte of interest (such as antibodies, aptamers, cells, enzymes); second the transducer, which transforms the analyte receptor bonding intensity into a distinct signal that an analyser can view; lastly the biosensor analyser, which presents the result in a user-friendly way. Biosensors for monitoring analytes in the blood, such as glucose, are now well established [25]. They are utilised in the early diagnosis of the lung [26], leukaemia [27], breast [28], colon [29] and other tumours [28]. The use of biosensors for early lung cancer diagnosis has been the subject of several review papers [24,30,31]. MRD detection and monitoring of treatment are new applications of the biosensor.

Although rapid detection of lung cancer based on liquid biopsy may overlap with treatment monitoring methods, in this paper, we focused on biosensors for liquid biopsy MRD detection in NSCLC. PCR-based methods (such as ddPCR, qPCR) and NGS for identifying biomarkers in lung cancer are not listed here. First of all, the MRD introduction in lung cancer is presented. Then the utility of biosensors in NSCLC, liquid biopsies and various samples are listed for this purpose. In addition, several types of biosensors for detection and monitoring are provided, along with a summary of the upcoming challenges.

## 2. MRD in Lung Cancer

MRD detection for conventional adjuvant drug decisions, personalised cancer monitoring in patients with early stage NSCLC and risk stratification for relapse in lung cancer patients is urgent (Figure 1). In NSCLC, various biomarkers have been proposed as MRD biomarkers, such as epidermal growth factor receptor (EGFR) mutations, K-ras mutations and excision repair cross-complementation group 1 (ERCC-1) expression, but further research is required before being utilised in the clinical trial. Liquid biopsy MRD detection may be considered as a possible prognostic marker in patients with NSCLC. Patients following treatment show using various techniques such as cancer personalised profiling by deep sequencing (CAPP-Seq) [32] and droplet digital PCR (ddPCR) [16] to provide evidence of MRD, which is associated with an increased risk of relapse. MRD identification utilizing ctDNA in radically resected NSCLC was linked to an increased risk of recurrence or mortality [33].

According to the NCCN guidelines list, *ALK* rearrangements, *BRAF V600E* point mutations, EGFR mutations, *METex14* skipping mutations, *NTRK1/2/3* gene fusions, *RET* rearrangements and *ROS1* rearrangements are key established predictive molecular biomarkers. In addition, blood PD-L1 expression, a key established immune biomarker, is significantly valuable. A high tumour burden has proven to correlate in response to programmed cell death PD-1 and PD-L1 inhibitors. The roles of *ALK*^+^ and *EGFR*^+^ in locoregional recurrence (LRR) and post-recurrence survival in NSCLC have been shown in multivariable analysis [34].

Therefore, it is recommended that patients be tested for these critical biomarkers after being diagnosed with metastatic NSCLC and ideally before they begin treatment. Effective targeted therapy or immunotherapy may be accessible depending on the results of the biomarker tests performed [2]. In addition, other biomarkers are actualised to detect and monitor lung cancer. For example, in many cancer types, including lung cancer, ENO1 (-enolase) expression is significantly correlated with reduced survival and a poor prognosis. Ho et al. described in 2010 an electrochemical disposable biosensor with a limit of detection (LoD) of 11.9 fg/mL that quantifies ENO1 [35]. Li et al.’s study showed that a label-free FET biosensor based on suspended single-crystalline graphene could be used for *ANXA2* (Annexin A2), *ENO1* and *VEGF* multiplex detection [36].

The clinical significance of MRD in lung cancer, as in most solid cancers, has been less well established, but studies have shown that liquid biopsy may be an ideal biomarker for assessing MRD. There are some clinical trials underway in cell-free DNA, ctDNA, EVs. The postoperative detection of ctDNA has poor prognosis in patients with surgically resected lung cancer [33,37]. The DYNAMIC clinical study showed that ctDNA analysis could provide a near-real-time read-out of tumour burden because the median ctDNA half-life was 35 min [38].

Target therapy and ICIs are limited to patients after surgery. However, target therapy and ICIs can be used for patients with detectable MRD. Detection of MRD by sensitive methods allows early therapeutic intervention. For example, the phase III MERMAID-1 clinical trial has recently demonstrated that MRD assessment could facilitate earlier, personalised adjuvant therapy for MRD^+^ patients, allowing for treatment escalation for these patients while minimising significant complications of adjuvant chemotherapy [39].

Routine morphological-based assays (imaging, biochemical, cytogenetic and immunological) cannot detect MRD. Hence, we need to introduce different molecular approaches that detect MRD before macroscopic relapse. Technique for MRD detection should have the following characteristics: (1) Be sensitive enough to detect at least one cancer cell in 1000 (in liquid-based test, this threshold is significantly lower compared); (2) be highly reproducible; (3) use of peripheral blood as its input sample; (4) cost-effective; (5) trustable to report the results quantitatively at different time intervals; (6) providing the simple data to interpret for the clinician. Although no common approaches have met all of these criteria to date, numerous strategies are employed for this goal. For example, Cobas EGFR mutation test v2 using plasma EGFR genetic testing for monitoring of disease progression of NSCLC received the FDA approval in 2016 to identify the mutation of exon 19 deletions and exon 21 substitutions throughout the time of fewer than 4 h.

The development of DNA sequencing technologies and biosensors has yielded promising results in liquid biopsy MRD detection. The detection of liquid biopsy MRD biomarker in the NSCLS is based on the following approaches: circulating tumour cells (CTCs), circulating tumour DNA (ctDNA) and EVs. Table 2 represents the list of various biosensors employed in the liquid biopsy MRD detection in lung cancer.

## 3. Liquid Biopsy as an Alternative Technique for Tissue Biopsy

Liquid biopsy is considered as a new technique for detecting tumour cells and identifying any tumour-derived products, which contains cell-free nucleic acids, circulating tumour cells and extracellular vesicles, specifically exosomes (Figure 2).

### 3.1. Circulating Cell-Free DNA (cfDNA) and Circulating Tumour DNA (ctDNA)

cfDNA (or ctDNA) monitoring is an optimistic technique for detecting MRD in tumours such as breast [61] and colon [62]. Natural cells release cfDNA into the biological fluids made by the ordinary activity of the cells, like apoptosis. Usually, the concentration of cfDNA is 5–10 ng/mL in body fluidics (with the range of 180–200 base pairs) [63]. However, the level of necrotic tumour cells (ctDNA) in cancerous tissue significantly increases [64]. cfDNA can present the heterogeneity of cancerous tissue regardless of any alteration during treatment or at different stages of cancer [65,66]. Preoperative ctDNA positivity in NSCLC advanced stage is a significant predictor of recurrence-free survival (RFS) and OS [63]. ctDNA identification after surgery increases the probability of early relapse detection and could play a critical role in stratifying patients for treatment [67]. Abbosh et al. demonstrated that phylogenetic ctDNA profiling tracks the subclonal nature of lung cancer relapse and metastasis. A unique procedure for ctDNA-driven therapeutic investigations proved a direct correlation between ctDNA concentration and computed tomography (CT) scan result of tumour mass [68].

### 3.2. Circulating Tumor Cells (CTCs)

CTCs or seeds of metastasis are isolated single migratory of cells or multi clusters of tumour cells which are measurable in the bloodstream of solid tumour cases. Similar to ctDNA, the lifetime of CTCs is 1~2.4 h, so instability of CTCs is a weakness of using this biomarker for investigation. However, the clinical application of CTCs has FDA approval (CellSearch^®^ platform) [69]. CTCs are analysed by quite common techniques like qPCR, NGS, cytogenetic analyses like fluorescence in situ hybridisation (FISH). Moreover, several methods are considered for isolation and detection of CTCs, such as antigen isolation based (EpCAM) on the surface of the CTCs, size-based and circulating tumour cell deformation. In personalised medicine, analyses of CTCs have the potential for monitoring patients treated with immunotherapy and chemotherapy and may become a new biomarker for early prognosis and MRD in NSCLC. Epithelial CTCs in the bloodstream of NSCLC cases are rarely discovered and aligned with a poor prognosis to checkpoint inhibitors [70]. Some studies demonstrated a similarity of approximately 93% between expression of PD-L1 on CTCs and PD-L1 expression in matched-patient tumours [71]. Patients with CTCs PD-L1^+^ had a shorter survival time than patients with CTCs PD-L1^−^ [72]. A day after surgery, CTCs numbers decrease to the lowest level, except for patients with confirmed recurrence, which increases after three days. These findings indicate that CTCs as a biomarker of biopsy fluid could be useful for monitoring MRD and responding to immunotherapy in early stage lung cancer after surgery [73].

### 3.3. Extracellular Vesicles (EVs)

EVs are bilayer biological vesicles found in extracellular fluids, including blood, urine and CSF. EVs are very complex in size, origin, content and function. In this regard, they are divided into different fractions and populations. The main classification of EVs could be divided into three main fractions based primarily on vesicle size (in diameter) and origin. These populations of EVs are exosomes, microvesicles (MVs) and apoptotic bodies. Exosomes are the smallest fraction or population among EVs. In contrast, MVs and apoptotic bodies are large in size and different in origin. Briefly, exosomes (30–150 nm in diameter) are originated from the multi vesicular endosomes (MVE) through exocytosis. MVs (150–1000 nm in diameter) are originated from cell cytoplasmic membrane through direct shedding of vesicles. The apoptotic bodies with heterogenous-sized vesicles (more than 1000 nm in diameter) are the cellular debris resulted from an apoptotic pathway. Besides the other vesicles’ populations, exosomes are more noticeable. Extensive research has been carried out for their biological and physiological functions in the human body, especially in specific conditions like cancers [74]. The vast potentials of exosomes and biological roles made them a new platform for different applications in biomedical sciences. Exosomes have a complex molecular content in their lumen, including RNAs, proteins, carbohydrates, lipids and metabolites. These exosomal contents belong to the cell that produces and secretes the exosomes. In this regard, exosomes harbour their parent-cell molecules that are different among various cells-secreted exosomes. The latter is the unique characteristic of exosomes (and MVs) that made them a source for biomarker discovery.

In the recent two decades, hundreds of biomarkers, so-called exosomal biomarkers, have been discovered and tested based on their sensitivity and specificity. These exosomal biomarkers are mRNAs, lncRNAs, miRNAs, proteins, carbohydrates and lipids. Due to the exosome’s vesicular nature, they are an excellent source for biomarker analysis versus non-exosomal biomarkers. Furthermore, exosomal biomarkers are resistant to harsh conditions and enzymatic digestion. So, they could be easily detected and analysed in different ways, including biosensors-based technology. However, technical aspects of exosomes purification and enrichment are challengeable from this point of view. Ultracentrifuge as a popular technique for exosome separation needs specialisation along with costly equipment. Low-cost techniques such as ELISA, Western blotting and flow cytometry require high concentrations of purified exosomes. As a consequence, purification, the isolation and detection of biomarkers on exosomes require innovative approaches.

## 4. Type of Biosensors for Detection of Lung Cancer Biomarkers in Liquid Biopsy

Biosensors, in the context of liquid biopsy MRD, based on bioreceptors, are classified into antibody, aptamer, DNA, enzyme, whole-cell and phage biosensors as well as based on the transducers are organised into electrochemical biosensors, electro-kinetic biosensors, optical biosensors and magnetic biosensors (Table 3). However, due to their remarkable properties, such as a biosensor’s limit of detection (LoD), wide linear dynamic range and excellent repeatability, they have gained popularity in recent years.

Currently, biosensors based on nanomaterials (nanobiosensors) are widely established in the development of biosensors due to their great sensitivity. Biosensors employ a variety of nanomaterials, including gold nanoparticles (AuNPs), metal oxides (MOs), nanowires (NWs), nanorods (NRs), carbon nanotubes-based biosensors (CNTs), quantum dot (QDs), graphene and its derivatives and nanocomposites (dendrimers) with the range size of 1 to 100 nm [75,76,77]. The popularity of gold nanoparticles (AuNPs) is because of the fast response and boost of analyte’s uncovering signals even in limited concentration [78]. Immobilisation of different biomolecules on the surface of carbon nanotubes (CNTs) makes them applicable; scaffolds having the ability to combine with a transducer signal. Layering structure of GR sheet makes two divisions of CNTs, single-walled carbon nanotubes (SWCNTs) and multi-walled carbon nanotubes (MWCNTs) which are practical for different applications [79].

### 4.1. Electrochemical Sensor

Cells and DNA/RNA as a target of liquid biopsy were sensed with label-free electrical biosensors. To enhance signal detection, lncRNA biomarker MALAT1 has been assayed by a DNA-based electrochemical biosensor as a target for lung cancer monitoring (Figure 3). The biosensor was developed based on a gold nanocage coupled with an amidated multi-walled carbon nanotube (Au NCs/MWCNT-NH2)-decorated screen-printed carbon electrode (SPCE). The result demonstrated that the LoD of the biosensor is 42.8 fM with a wide linear range of 10^−7^–10^−14^ M [40]. The existence of cytokeratin 19 fragment (CYFRA 21-1) in serum is considerably matched up with NSCLC and may be able to predict tumour relapse [80].

Sandwich-type immunosensor, employing immunosensor amino-functionalised carbon nanotube (MWCNT-NH2) within high efficiency in electrical conductivity and electron transfer and suitable biocompatibility, makes this nanotube an agreeable detector for abnormal expression of CYFRA21-1 in serum [81]. Assessment of CYFRA 21-1 has been identified as a potential prognostic factor in NSCLC. Studies showed that serial measurement of CYFRA 21-1 in patients with the advanced NSCLC-undergoing treatment and the impact of immune-checkpoint inhibitors (ICPIs) could be used to assess the outcome of early treatment without the need for frequent and expensive imaging procedures. It is a valuable marker to detect recurrence during monitoring after surgical resection [82]. Yoon et al. established the functionalised graphene oxide nanosheets chip for isolation of CTCs from early stage lung cancer patients’ moreover adjustable for advanced metastatic cancer patients. Graphene oxide nanosheets modified by phospholipid–polyethylene–glycol-amine (PL–PEG–NH_2_) and tetrabutylammonium (TBA) hydroxide were adsorbed onto the patterned gold surface and then chemically conjugated with NeutrAvidinis, which is the connection between crosslinker on graphene oxide nanosheets and EpCAM antibodies [83].

The electrochemical cyto-sensors (PDA NPs nanocomposites interface on the functionalised PGE) with high biocompatibility and sensitivity have been designed by Bolat et al. for label-free electrochemical detection of A-549 lung cancer cells in a wide linear range (1.0 × 10^2^–1.0 × 10^5^ cells mL^−1^) with a low detection limit (25 cells mL^−1^). In this regard utilising electrochemical cyto-sensors for bio-detection of cancer cells in liquid biopsy makes them a fast and straightforward sensor for bio-recognition elements to investigate the capture and detection of lung cancer cells in early stages [84]. Fusion of a self-assembled monolayer (SAM) of AuNP and determined aptamer on gold microelectrodes for target cells trapping fabricated a biosensor with high quantification stability operable for early lung cancer detection. The S–Au bonds between thiol-terminated aptamers and AuNP layers showed considerable selectivity assay for A549 lung CTCs cells capturing in the whole blood samples [85].

Commercialisation and mass production of user-friendly biosensors for liquid biopsy has been expanded. Electric field-induced release and measurement (EFIRM) liquid biopsy can identify ctDNA in patients with early stage NSCLC. An electrochemical control system with 96 parallel channels drives the EFIRM platform. This electrochemical control system is linked to a gold electrode array; the size of a standard microtiter plate. In an effort, EFIRM biosensors, which are effective, accurate, fast, user-friendly and inexpensive, are developed for non-invasive saliva-based EGFR gene mutation detection [86]. The EFIRM-based method was also used in the detection of the prognostic biomarker in liquid biopsy such as ultra-short circulating tumour DNA (usctDNA) in plasma and saliva [87] and p.L858R EGFR, exon19 del EGFR, p.E545K PIK3CA ctDNA in plasma [88] and exosomes [43]. The saliva-based EFIRM platform shows good concordance to plasma-based platforms (ddPCR and NGS) for longitudinally monitoring the combination of EGFR and PIK3CA ctDNA and may be a valuable platform for liquid biopsy MRD in NSCLC patients. EZLife Bio has developed a device based on the EFIRM technology that takes only 3 h to run EFIRM assays, which is substantially faster than NGS-based liquid biopsy methods. In many investigations, the pH value of the intracellular fluid environment of cancer is ≤7.1 [89]. The correlation between pH and cancer can be indicated as a biochemical marker for diagnosing and monitoring NCSLC cancer. Detection of CTCs in the blood either with deterministic lateral displacement (DLD) or without DLD devices by using the potential difference between the reference electrode (Ag/AgCl) and indicator electrode (ZnO) in a microfluidics-based pH sensor is reported by Ganesh Kumar Mani et al. [90].

### 4.2. Electro-Kinetic Biosensors

The pressure-driven fluid flow transport by advection enforces both a current and a potential at the charged micro-and nanochannel ends. This leads to an electrokinetic phenomenon that can be used as a sensor/transducer principle. The generated electricity is well-regulated as a sensor power supply, so this effect is useable as a passive sensor/transducer principle. In this way, the adsorption of biological material like biomarkers and physical changes like pressure/flow or pH value is detectable [91]. Recently, a multiplexed electro-kinetic sensor was used to monitor several small extracellular vesicles (sEVs) and tumour cell surface markers in sEVs isolated from pleural effusion (PE) fluid of NSCLC patients [92]. The detection method is based on the electro-kinetic principle, where the change in the streaming current is monitored as the surface markers of the sEVs interact with the affinity reagents immobilised on the inner surface of a silica microcapillary. This study obtained samples from patients with diverse genetic drivers of their cancers, including *EGFR*, *ALK* and *KRAS*. They were also given different treatment regimens, including PD-L1 immune checkpoint inhibitors, EGFR- and ALK-TKI. In NSCLC patients treated with EGFR-TKIs, PD-L1 expression is a poor prognostic factor. The biosensor measured the PD-L1 expression in sEVs. It showed that an electro-kinetic biosensor could detect various and simultaneous quantities of low concentration biomarkers in pleural fluid and predict or monitor therapy response. The electro-kinetic results were consistent with those obtained using conventional methods such as Western blot (WB) and proximity extension assay (PEA). Although the outcomes of this biosensor seemed acceptable, it is possible to enhance the signal of proteins by modifying the size and charge of the particles in the electro-kinetic sensor [93]. A microfluidic cathodic photoelectrochemical immunosensor for measuring CYFRA 21-1 has been developed. The process is based on the p–n junction of AgI/Bi_2_Ga_4_O_9_, with dissolved O_2_ acting as an electron acceptor to create the superoxide anion radical (O_2_^−^) as the working microelectrode. When combined with a novel SOD@hMnO_2_ as the co-catalyst signal amplification label, O_2_ can be catalysed by SOD via a disproportionation reaction to produce O_2_ and H_2_O_2_; then, hMnO_2_ can trigger the decomposition of H_2_O_2_ to produce O_2_ and H_2_O_2_. The linearity of the microfluidic cathodic PEC biosensor chip ranged from 0.1 pg mL^−1^ to 100 ng mL^−1^, with a detection limit of 0.026 pg mL^−1^ [46].

### 4.3. Magnetic Biosensor

Whenever the magnetic field is enforced to the sensor, the resistance of giant magnetoresistance (GMR) sensor changes; as a result, a biomolecule which is labelled with magnets, generates a signal. High sensitivity and rapid response induced by a low magnetic field are the superiority of GMR material for biosensors. Furthermore, the stability of magnetic probes for labelling compared to other labels like fluorescent tags for long-time usage, external manageability, and no background noise are some advantages of magnetic biosensors [94]. For signal amplification, magnetic particles were employed in electrochemical biosensors [30].

Nair and colleagues measured the tissue inhibitor of metalloproteinase-1 (TIMP1) biomarker with the help of the immunoassay GMR sensor [95]. In this study, plasma TIMP1 levels and relevant clinical and imaging characteristics were adequate for diagnosing lung cancer in smokers. In a 2011 research, Pesta and colleagues demonstrated the prognostic effect of TIMP1 mRNA. Higher TIMP1 mRNA levels are associated with an adverse prognosis of patients [96]. The expression of TIMP-1 may serve as a prognostic biomarker as well as a therapeutic target in NSCLC patients [97]. It is assumed that TIMP-1 is a possible tissue biomarker for lymph invasion and distant metastasis of lung adenocarcinoma by promoting cancer metastasis [98].

### 4.4. Optical Biosensors

Optical biosensors are a compressed analytical device that takes advantage of the principle of optical measurements (absorbance, fluorescence, chemiluminescence and others). The yields of optical biosensors are signals which present the concentration of the analyte. Direct, real-time and label-free detection of substances are some of the optical biosensors’ benefits, along with high specificity and sensitivity. Generally, reference sensors are not required during the measurement since the comparison signal can be generated with the same light source as the scanning sensor [99].

#### 4.4.1. Surface Enhanced Raman Spectroscopy (SERS)

SERS-based biosensors are divided into direct (label-free) detection and indirect detection (needing SERS tags). Label-free presents molecular fingerprint information of biomolecules. The drawback of label-free detection is the complexity and even weakness of signals from some substances; also, they are prone to signals from interfering molecules in the matrix. In indirect SERS sensing, unique SERS signals (SERS tags) are labelled on the intensifying substrates. SERS read-out is recognised after bounding targets and ligands or the reaction of SERS tags to environmental properties [100].

Exosomal PD-L1 in the blood is a good predictor of clinical response to anti-PD-1/PD-L1 therapy. Pang and colleagues applied the Anti-PD-L1 antibody modified Au@Ag@MBA (as the SERS tag) for exosomal PD-L1 labelling and quantification [101]. One of the most intriguing aspects of this biosensor was separating exosomes based on Fe_3_O_4_@TiO_2_ nanoparticles without antibodies or aptamers, which can enrich and separate exosomes from solution in 5 min with a 96.5 percent capture efficiency. In addition, TiO_2_ can specifically bind with the phosphate groups on the lipid bilayer of exosomes. This investigation showed that SERS-based biosensors could be utilised to monitor immunotherapy in the blood [102].

In another study, an ultrasensitive sensor based on an Ag nanorod array SERS substrate by assembling special hairpin-shaped molecular beacons (MBs) was developed for the detection of multiple lung cancer-related miRNA biomarkers [103]. Molecular beacons a hairpin-shaped 25 nucleotides DNA or RNA molecules with a fluorophore and a quencher. They do not fluoresce in the absence of their target complementary molecules. Instead, a spontaneous configuration change on MBs occurs due to hybridisation with their particular targets, and the dye and quencher separate from each other, resulting in fluorescence emission.

#### 4.4.2. Surface Plasmon Resonance Biosensors (SPR)

Different interactions simultaneously with high throughput and sensitivity are characteristics of plasmon surface resonance imaging (SPRi). Spatially label-free optical detection and analysis of bio interaction in multiplex format make this technique an excellent detector for identifying many parameters simultaneously. The SPR technique could be used to identify the characteristics of NSCLC-related exosomes. Fan et al. recently developed a novel SPRi-based biosensing technique for detecting NSCLC-derived exosomes via multiple recognition sites with LoD of 10^4^ particles/μL. The number of plasma exosomes identified with Anti CD63 differed across patients in a study of four healthy samples, four treated samples and four patients. Exosome levels were lower in healthy participants (approximately less than 10 RU) and higher in patients (about 25 RU). For patients undergoing therapy, intermediate values of the two groups were recorded. These findings suggest that immune-base sensing assay can be employed to assess treatment effectiveness by assessing the abundance of exosomes in plasma [50].

Some biosensors detect cargo within exosomes, such as circulating tumour RNAs (ctRNAs). Recourse to exosomes in blood and urine samples for early identification and monitoring treatment for various cancers such as prostate [104] and AML [105] has been reported.

Detection of MRD biomarkers in leukaemia has revealed that using several markers improves the accuracy and reliability of the MRD approach considerably [106]. As a result, seemingly using multiplex approaches in the diagnosis of MRD is desirable. Exosomal miRNAs have been proven to be useful as promising diagnostic biomarkers in patients with NSCLC. Since their remarkable stability and abundance, exosomes are better adapted for liquid biopsy than CTCs and cfDNA.

In the light of this, exosomal miRNAs have been considered as possible biomarkers for recurrent lung cancer [107]. However, information is scarce in this field, and specific conflicts must be resolved [108]. An SPRi-based biosensor was designed to investigate the simultaneous detection of multiplex 24 NSCLC-associated exosomal miRNAs in a clinical sample employing Au-on-Ag heterostructure and DNA tetrahedral framework (DTF) [49]. This biosensor has a detection range of 2 fM to 20 nM and a detection limit of 1.68 fM. In addition, DTF immobilisation on the gold array chip minimises nonspecific adsorption, resulting in fewer false-positive results due to interference factors in the input sample.

Liu et al. developed a highly sensitive, compact SPR biosensor to capture exosomes and characterise exosomal proteins. At concentrations as low as 2 × 10^10^ exosomes/mL, a compact SPR biosensor successfully detected EGFR expression levels in A549 exosomes, two times higher than the ELISA method (4 × 10^10^ exosomes/mL). Furthermore, there were no significant differences in exosomal EGFR expression in the clinical phase between the serum sample of the healthy and cancer patients [52], so this must be considered.

#### 4.4.3. Fluorescence-Based Biosensing

Label-free and fluorescence-based/luminescent are two detection techniques applied in optical immunosensors. Fluorescent-based biosensors result in read-out and requires expensive devices while being highly sensitive. While label-free biosensors are cheap, they do not have acceptable sensitivity. Thus, the false results of these biosensors limit their clinical use. The great sensitivity of fluorescence detection combined with ligand-binding proteins is among the most prevalent techniques in optic biosensors. Biosensors are utilised with fluorescent labels or tags, including carbon nanotubes, nanoparticles, quantum dots, nanowires and nanorods [109]. The antigen/antibody is attached with fluorescent labels in the fluorescence detection method, which causes a change in the fluorescence intensity, indicating the presence of the target biomarker. The degree of fluorescence emitted correlates to the magnitude of the interaction between the analytes and bioreceptors.

All in all, minimal regent consumption (and input sample such as blood, urine or purified exosomes) microfluidic-based nanobiosensors offer a high automation capability, a fast reaction time, cost-effectiveness. Bai et al. described a novel microfluidic immunoassay system for exosome isolation and multiplex detection using beads queued uniformly at gaps among the microarray [110]. QDs have broad excitation and narrow emission spectra [111]. The use of QDs for immunoassay detection improved fluorescence stability and multiplexed biomarker measurement. Exosomes that were isolated in that sensor utilise Anti-CD9-labeled magnetic beads. Three QD probes labelled with tumour marker antibodies were employed for multiple detections of CEA, CYFRA 21-1 and progastrin-releasing peptide (ProGRP). This study demonstrated that these biosensors could distinguish plasma-derived exosomes from lung cancer patients and healthy controls. Furthermore, in patients with SCLC who had higher ProGRP levels upon diagnosis, a drop in ProGRP levels may be a good predictor of objective response to chemotherapy, according to a study published in 2020 [112]. In addition, ProGRP is a reliable biomarker for detecting SCLC and distinguishing it from NSCLC [113].

Graphene oxide’s (GO) fluorescence quenching property has been proven and utilised for fluorescence imaging and biosensing. A novel graphene oxide (rGO)-magnetic silicon microspheres (MNPS) and DNase I nanosensor based on FRET to detect serum PD-L1 was developed [114]. The biosensor can detect PD-L1 with a linear range from 100 pg mL^−1^ to 100 ng mL^−1^, with a detection limit of 10 pg mL^−1^. This fluorescent sensing method provides promising application in prognosis biomarker detection.

High expression of miR-155 and miR-22 predicts recurrence in NSCLC [115,116]. Recently, a homogeneous T7 exonuclease (T7 Exo)-assisted signal amplification combined with GO quenching platform has been applied to detect miRNA-21 and miRNA-155 simultaneously with FAM and ROX-labelled single-strand DNA probes involved in drug resistance [117].

#### 4.4.4. Lateral Flow Immunoassay (LFIA)

LFIA immunoassays are a versatile tool in the field of point-of-care (POC) diagnostics. Routine LFIA test performs through antibodies. As an alternative, the utilisation of aptamer bioreceptors instead of antibodies in immunoassays has shown promising results. Aptamer provides several advantages over antibodies, including a shorter production process, high-throughput scalable sensor production and excellent stability and reproducibility [118]. LFIA immunoassays using AuNPs as labels successfully isolate lung cancer-derived exosomes. Recently, a lateral flow aptamer assay (LFAA) strip, based on an aptamer against CD63 protein on exosome membrane, has successfully isolated exosomes derived from human lung carcinoma cells [58]. Low CD63 expression may indicate a poor prognosis for patients with NSCLC [119].

Despite the meagre cost of these tests, they are less sensitive. On the other hand, due to the qualitative or semi-qualitative nature of LFIA-based tests, it is expected that they will not be considered as a sensitivity method for MRD in clinical practices. Some attempts have been made to address these challenges. For example, a magnetic lateral flow immunoassay has been recently developed for small extracellular vesicles quantification [120].

#### 4.4.5. Chemiluminescence

Electronic luminescence (ECL) sensors are composed of electrochemistry and visual luminescence measurements. When a potential is applied to an electrode, excitation of the electrode surface occurs. Thus, electron transfers among molecules are happed, and light emission is subsequently measured. Luminescent signals are produced by ECL biosensors when unique biological interactions recognise an analyte.

Wang and colleagues recently introduced a sandwich-type electrochemiluminescent biosensor for sensitive electrochemiluminescence detection of CYFRA21-1 in the cell. This is the first attempt to build an ECL biosensor based on CD-MOF@Ru (bpy) ^2^_3_^+^ modified on the glassy carbon electrode (GCE) [121].

### 4.5. Other Biosensors

Ramanathan and colleagues have demonstrated good EGFR mutation detection for NSCLC utilizing an unsophisticated, affordable nanosensor with an aluminosilicate nanocomposites-enhanced sensing area. Because of the large surface area, aluminosilicate nanocomposites improve DNA probe attachment to the biosensor. The calculated LoD of the aluminosilicate modified genosensor is 100 aM. In addition, a voltammetry interdigitated electrode (IDE) was developed to sensing genomic DNA hybridisation using microscale aluminium electrodes and gaps [122].

## 5. Challenges and Future Prospective of Liquid MRD

Biosensors transform target analyte detection into a quantifiable digital signal that may be used to detect various lung cancer MRD biomarkers in liquid biopsy. The emergence and advancement of micro and nanofabrication technologies open up a new vista for fabricating sensitive, efficient and cost-effective biosensors for the clinical setting. Shortly, innovative POCT biosensors will be developed for MRD monitoring in cancers, including lung cancer. These ultrasensitive sensors can detect trace quantities of target biomarkers. These integrated technologies have great promise to overcome the limitations of current lab-based diagnostics and provide personalised POC testing for lung cancer.

Reliable MRD diagnosis is employed to predict relapse, risk classification, treatment selection and guide risk-adapted adjuvant immune therapy. Data from multicentre prospective studies in other tumours such as colon [9] and breast cancer [123] have shown that precision medicine largely relies on MRD. An appropriate drug dose is administered to patients through MRD detection, therefore preventing the drug’s adverse effects. Moreover, liquid-biopsy MRD can be considered as a surrogate factor in the development of novel drugs.

There are several challenges in determining MRD with bio-sensors in solid tumours, and any technique used for MRD should address these challenges. Current MRD biosensors have some drawbacks like the requirement for clinical approval, testing on large samples, the lack of sensitivity and specificity, cost, automation and standardisation. However, because of the ease of access to target tissue in MRD monitoring, the limitations of identifying tumour-associated mutations appear to be minimised. In addition, the aforementioned technique reduces the number of false positives.

Over the past few years, integration of high-throughput microfluidic methods such as lab-on-chip devices can minimise the size of bio-sensors. They furthermore resolve some of the previously mentioned limitations. In addition, it removes some hurdles to simultaneous measurement of biomarkers. Priorities of microfluidic devices such as automation and high-speed analysis of patients’ liquid biopsy samples make them an advanced approach [124]. Furthermore, recent advances in microfabricated devices have significantly contributed to enabling high throughput analysis. For example, a microfluidic device with an array electrode is designed to test 28 samples simultaneously in <2 h [60].

Approximately 94% of stage II–III patients with recurrence had ctDNA in blood. MRD detection after treatment of localised lung cancer would be effective. It may be possible to provide personalised adjuvant treatment when the disease burden is the lowest [32]. The in vitro assay identified the phylogenetic subclone involved in recurrence in a wide range of patients. The MRD/recurrence can be detected weeks to months before imaging (median 70 days) [68]. Cancer recurrence appears after adjuvant therapy in some early stage NSCLC cases. Liquid biopsy biomarkers detection by biosensors may allow for the personalisation of adjuvant therapy based on the genetic profiling in patients; hence, the side effects of treatments may be controlled and prevented. High-sensitivity biosensors can detect ctDNA and other MRD biomarkers. Prospective clinical studies are necessary to validate the scenario.

In the adjuvant therapy NSCLC context, low-frequency mutation detection in plasma cell-free DNA can be utilised as a patient stratification tool. Monitoring several mutations per patient improves platform sensitivity for ctDNA detection in an MRD context. In this regard, efforts should be directed toward developing a novel nanoparticle with an integrated microfluidic device (as a lab on chip device) for performing simultaneous detection of numerous lung cancer MRD biomarkers in order to deal with the diverse characteristics of lung cancer, particularly in phases II–III of NSCLC patients, as well as following ICI therapy. Signal amplification is often needed to detect low abundance DNA/RNA or proteins biomarkers in the biological fluids [125]. A catalytic hairpin DNA circuit (CHDC) is an enzyme-free DNA circuit, which facilitates signal amplification by the catalysed hairpin assembly. This approach has the potential to be a low-cost, fast and sensitive tool for detecting considerably lower quantities of ctDNA in blood. Integrating biosensors with CRISPR-based nucleic acid diagnostics is a cost-effective way to measure biomarkers such as mRNAs and miRNAs. Recently, a chip called COMET has been developed using a CRISPR/Cas13a system and the CHDC technology with a reusable electrochemical biosensor for the rapid detection of multiple RNAs [126]. Although biosensors’ sensitivity to identify and quantify biomarkers is adequate, further steps are required to define the accuracy of liquid biopsies as a necessity for clinical tests.

Preparation, characterisation and separation of EVs, CTC and cfDNA are some of the difficulties besides low concentration of biomarkers, particularly in the early stages of cancer, which exacerbates these issues, as well as preparing samples are time-consuming and costly with current technologies. Microfluidic-based methods show a good application in the isolation of CTCs for clinical usages. A valuable approach to detect heterogeneity of CTCs in NSCLC is using the automated platform LiquidBiopsy^®^ armed by the high throughput sheath-flow microfluidics chip. It was coated with anti-EpCAM antibodies as a functionalised surface to detect clinically relevant genetic profiling of CTCs populations. Thereby CTCs populations are accessible for molecular and direct automated DNA analysis [127].

The other challenge is target biomarkers; biomarkers for risk stratification can serve as tools for evaluating lung cancer patients and selecting the best adjuvant treatment. In addition, biomarkers will play an increasingly important role in lung cancer’s progression, and sensitive and reliable biosensors for clinical decision-making will become more readily available as lung cancer pathophysiology is better understood. Biosensors are outstanding at detecting multiple biomarkers.

MRD’s Achilles heel is time. Although selecting a biomarker and understanding how to quantify it is critical, it is not adequate. However, no appropriate timing for MRD examination has yet been established. MRD can be performed post-resection in early stage NSCLC or early stage NSCLC based on liquid biopsy. Although the ELN has established guidelines for leukaemia and lymphoma, liquid biopsy time is one of the most significant issues facing solid malignancies. Efforts in this area have proven fruitful. Prospective DYNAMIC research was conducted in 2019 to investigate perioperative dynamic changes in ctDNA in patients with primary lung cancer. According to the findings of this investigation, ctDNA decays rapidly following tumour removal. Three days following surgery can be used as the baseline value for post-surgical MRD lung cancer surveillance and is useful for informing clinical decision-making [128].

Hematopoietic stem cells (HSCs) can bear recurrent somatic mutations of leukaemia-associated genes that can be seen in a healthy individual (clonal haematopoiesis of indeterminate potential, or CHIP mutation). These mutations contain somatic mutations but are not associated with malignancy [129]. CHIP-derived mutations are present in late-stage NSCLC [130]. CHIP expression should be considered when interpreting blood-based liquid biopsy results since it can lead to false-positive results. Blood is commonly used in liquid biopsies. Other fluids, such as urine, can be used as an alternative sample to address CHIP-based challenges. Moreover, urine samples appear to have lesser protein levels, which reduce false positives and other interfering factors. So, biosensor based on other biological fluids such as urine and sputum [131] may improve the performance of routine biosensor assays. Colli-Pee^®^ has created a novel technique for non-invasive urine collection and stabilisation in this regard. In lung cancer, urinary circulating DNA profiling demonstrated that urinary DNA measurements could potentially be helpful for disease monitoring of MRD in NSCLC [67]. Detection of specific biomarkers in different stages of NSCLC by biosensors is a complementary procedure to standard techniques that might be considered a supportive approach for determining the most appropriate treatment for each patient as well as early detection of recurrence and metastasis.

## 6. Conclusions

Biosensor-based test for liquid biopsy MRD and treatment monitoring has shown bright horizons in NSCLC. A lower concentration of ctDNA and CTCs in body fluids can be detected by a biosensor. The widespread use of biosensors in MRD diagnosis reduces costs and expands the availability of these tests to monitor treatment or choose adjuvant immunotherapy in early stages of patients. We expect biosensing-based MRD detection technology to be built into micro- or nano-structured platforms with unprecedented accuracy and reliability in the next decade.

## Figures and Tables

**Figure 1 biosensors-11-00394-f001:**
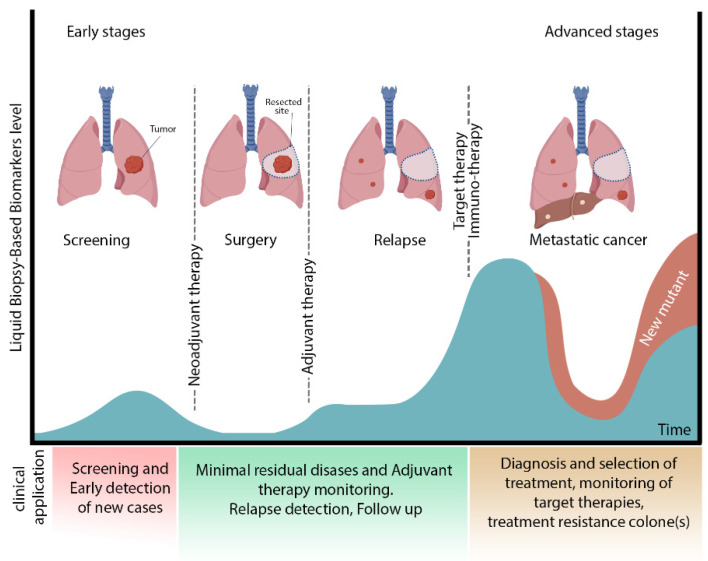
Different stages of lung cancer and clinical application of liquid biopsy. This figure shows the different stages of lung cancer. After surgery, the presence of MRD induces a recurrence. Immunotherapy or target therapy is recommended for treatment and management of relapse. Biosensors are functional for early detection at the screening stage besides identifying new mutants in the advanced cancer stage. Future aspects of biosensors are the assistance of select comprehensive therapy (adjuvant therapy) and monitoring lung cancer at different levels.

**Figure 2 biosensors-11-00394-f002:**
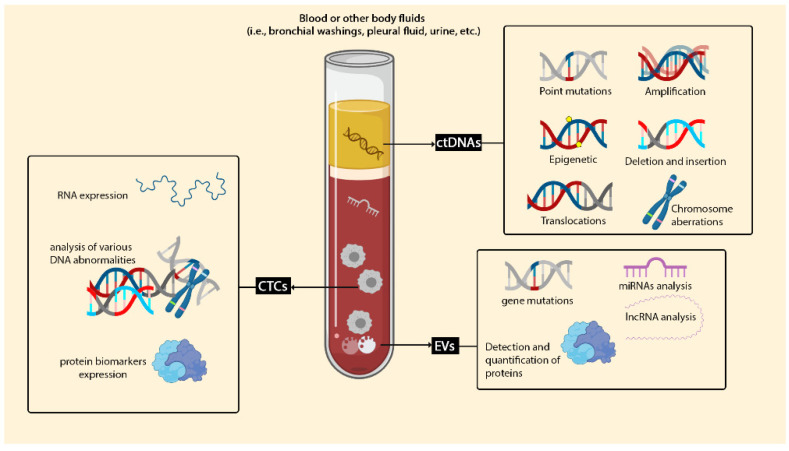
Liquid biopsy in lung cancer. Liquid biopsy can be considered for plenty of clinical targets. Liquid biopsy biomarkers are categorised into three main groups: circulating free DNA (cfDNA), CTCs and EVs. cDNA can be used for detecting a wide variety of mutations, such as insertions, deletions and amplification. Capturing and identifying CTCs in whole blood are cooperative for checking proteins and RNA expression and analysing various DNA abnormalities.

**Figure 3 biosensors-11-00394-f003:**
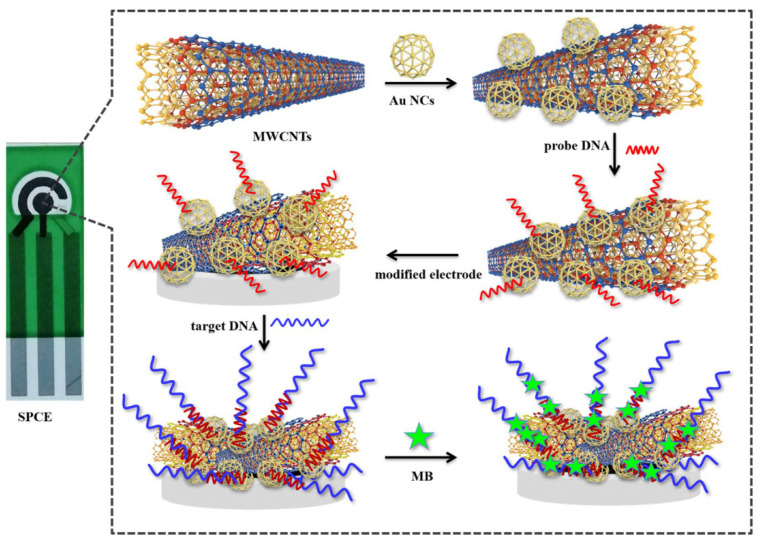
Schematic representation of the SPCE electrochemical DNA biosensor to detect lncRNA biomarker MALAT1. Reprinted from ref. [40].

**Table 1 biosensors-11-00394-t001:** Advantages and disadvantages of routine MRD methods in clinical.

	Principle of Method	Advantages	Disadvantages	Application in MRD of Lung Cancer
Traditional Biopsy and morphological-based test	A piece of lung tissue is taken from the body for histopathology evaluation.	Clinically validated	Invasive and risky. Unpleasant for the patient. Inter and intra laboratory error during reporting of the result. Failure to detect metastasis in other tissues. Serial sampling is challenging. Low sensitivity.	Traditional methods for MRD detection. Immunohistochemistry staining to identify and monitor biomarkers
ddPCR	Digital PCR (dPCR) divides a bulk PCR reaction into millions of nanolitre-scale microreactions, each containing zero, one, or only a few DNA molecules. Absolute quantification of the material by dPCR is accomplished by counting positive nano responses and using Poisson statistics.	Costly; ultra-sensitive; no need for a calibrator for quantification. Absolut quantification; fast.	False-positive and negative results. Requires a skilled technician. Not routinely available in many clinical laboratories. Sophisticated instruments are required	It is an ultrasensitive method for detecting pre-defined mutations. MRD quantification and treatment monitoring are possible without the need for a calibrator curve.
NGS	Next-generation sequencing (NGS) is a massively parallel sequencing technology to large-scale DNA sequencing.	Applicable to all known and new mutations.	Well-trained technicians are needed. Analysing the results is time-consuming. Costly, requires high DNA input, low sensitivity, rapid turnaround times.	Unlike other methods, it is possible to analyse at the genome-wide level.
Biosensor		Simple; ultra-sensitivity; disposable test. Applicable as a POCT test.	Clinically not well validated. False result	As a new and low-cost method, it can perform tests at consecutive times. Different bioreceptors can detect MRD on liquid biopsy. As a POCT test, it is possible to perform the test at the bedside.

**Table 2 biosensors-11-00394-t002:** A classification of biosensors for lung cancer monitoring.

Detection Method	Sample	Biomarkers	Biosensor	Limit of Defection (LoD)	References
Electrochemical	Synthetic lncRNAs	MALAT1	SPCE	42.8 fM	[40]
	Saliva and plasma	EGFR L858R and exon 19 del mutations	EFIRM	NA	[41]
	Small extracellular vesicles (sEVs)	EGFR	Electrokinetic Sensor	2.8 × 10^8^ particles/mL	[42]
	Serum and saliva	-	EFIRM	NA	[43]
	Synthetic DNA	EGFR	sandwich-assays	NA	[44]
	Serum	CYFRA21-1 and CEA	3D graphene (3D-G), poly-thionine (pThi) and poly-m-Cresol purple (pMCP)	0.18 ng/mL (CYFRA21-1) 0.31 ng/mL (CEA)	[45]
	Serum	CYFRA21-1	Microfluidic	0.026 pg mL^−1^	[46]
	Serum	CYFRA 21-1	ECL and eATRP signal amplification	0.8 fg mL^−1^	[47]
	Exosomes	EGFR and PD-L1	Electro-kinetic	4.9 × 10^6^ particles/mL	[48]
SPR	Exosomes miRNAs	miRNA-21, 378, 200, 139	SPR	1.68 fM	[49]
	Exosomes	Anti-EGFR and anti-EpCAM	SPRi	2.37 × 10^4^ particles/μL	[50]
	Serum (Protein)	ALCAM, TAGLN2	SPR imaging sensor with polarisation contrast	6 ng/mL (ALCAM) 3 ng/mL (TAGLN2)	[51]
	Exosomes	EGFR, PD-L1	Nanoplasmonic exosome (nPLEX) assay	9.258 × 10^3^%/RIU	[52]
SERS	Serum	Cytochrome c (Cyt c)	Aptasensor	1.79 pg/mL (Serum) 1.148 pg/mL (PBS)	[53]
	CTCs	EpCAM	antibody-adsorbed nitrocellulose membrane	NA	[54]
Fluorescence	Tumour-derived exosomes (Plasma)	IGF-1R	Microfluidic device	0.28–0.38 pg/mL	[55]
	cell extracts	CDK6 Kinase	fluorescent peptide biosensor	NA	[56]
Magnetic	CTCs	EGFR	Immunomagnetic and Magnetic Sifter	NA	[57]
Aptamer	Exosomes	Identification of A549 exosomes	lateral flow aptamer assay	6.4 × 10^9^ particles/mL	[58]
Micronuclear magnetic resonance	CTCs	EGFR, EpCAM, HER-2, MUC-1	μNMR	NA	[59]
colorimetric	Serum	Monitoring soluble immune checkpoints (PD-L1, PD-1), (LAG-3)	microfluidic sandwich immunoassay (multiplexed immune checkpoint biosensor (MICB)	5 pg mL^−1^ (PD-1 and PD-L1) 50 pg mL^−1^ (LAG-3)	[60]

**Table 3 biosensors-11-00394-t003:** Comparison of different biosensors used to diagnose MRD liquid biopsy.

Transduce	Principle	Advantages	Disadvantages	Application in Lung Cancer MRD
Electrochemical sensor	Convert the biochemical interaction to electrical signals	- User-friendly - Portability - Cost-effective - Easily miniaturised - Rapid - High-specificity - High-reproducibility - Low detection limit - Need a small sample volume - Label-free - Real-time detection (piezoelectric) - Can be performed in turbid samples	- Low sensitivity - False result from nonspecific bonding - Low shelf life - Cells or biomarkers are no longer intact. - Sensitive to temperature or environment change implications	- Easily miniaturised, making them ideal for personalised medicine in the MRD detection context. - Electrochemical aptasensors can be reused; therefore, a biosensor can be used for serial monitoring at different times. Aptasensors have lower sensitivity than other electrochemical methods. - Capable of detecting ctDNA at extremely low quantities. - Able to detect specific ctDNA mutations in untreated serum or blood - Sequencing results are provided faster than standard PCR and DNA techniques.
Magnetic biosensor	Applying paramagnetic particles to detect biological interactions by monitoring magnetic property changes	- Low background noise - Detection of multiple biomarkers at the same time - High sensitivity - Can be performed in turbid samples - Stability - No change in the nature of magnetics in response to chemical reagents	- Costly - Need to multiple washing steps - Time-consuming	- For quantitative MRD testing, magnetic nanoparticles used with other bio-sensing platforms such as LFIA - Magnetic tags apply to the isolation and characterisation of EVs and CTC, so no sample preparation is required in an integrated biosensor. - Application as an immunoassay-based liquid biopsy or DNA-based liquid biopsy or POC devices - NMR signal detection needs only a few sample purifications steps, making it ideal for developing liquid biopsy for low-quality fluid samples or those with excessive protein contamination.
Surface-enhanced Raman spectroscopy (SERS)	Using molecules adsorbing on rough metal surfaces to generate Raman scattering	- High spatial resolution - Non-invasive label-free - High sensitivity - Multiple detections - Quantification - Unique spectroscopic fingerprint - Low background noise	- Complex structure - Requires a nanoprobe - -Time-consuming - Expensive equipment to read-out - Requires a trained technician - Non-stability in long-term storage	- SERS tags have extraordinary multiplexing capacity. - Real-time monitoring of therapy response
Surface plasmon resonance biosensors (SPR)	Alteration of SPR angle due to increasing of refractive index during binding of biomolecules on the sensor surface	- High sensitivity - Cost effectivity - Real-time detection - Reliable - High sensitivity - Enzyme-free - Label-free - Stability	- Complicated optics - Costly - Low sensitivity	- Monitoring of immune checkpoint inhibitors (ICI)
Fluorescence-based biosensing	An analytical signal of a photoluminescence emission mechanism	- High photostability (quantum dots) - High sensitivity - Imagining technique	- Requires a trained technician - Background fluorescence noise - Sample losses during the labelling and purification process - Costly	- Monitoring of ICIs such as PDL1 - QDs are used as energy donors in FRET for the CTC identification as a prognostic factor - Due to high sensitivity, reliability and reproducibility, fluorescence techniques are favourable for detecting MRD biomarkers than other optical methods.
Lateral flow immunoassay (LFIA)	Optical properties	- Rapid - Portable - User-friendly - Cost-effective - Additional processing or external equipment is not required - No need for skilled personnel - Long shelf life - No need for additional equipment	- Qualitative or semi-quantitative read-out - Low sensitivity	- LFIA test strips are handy for detecting EVs, especially when a rapid procedure is needed. - They have the potential to become a quantitative test. - For quantitative MRD testing, gold or latex nanoparticles are replaced with magnetic particles or quantum dots (QDots-based lateral flow test strip).
Chemiluminescence	Electrochemistry and visual luminescence measurements	- Low background noise - Simple instrumentation - High sensitivity - Broad dynamic range - No need for the external light source	- Background noise - Reagent low stability - Time-consuming - Sample vanishing during the labelling and purification process	- A shorter DNA region is required. Compared to other routine methods such as PCR-based tests, DNA with a short length (~18 nucleotides) is detectable.

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
