# Peer review of "Liquid Biopsy-Based Biosensors for MRD Detection and Treatment Monitoring in Non-Small Cell Lung Cancer (NSCLC)"

_biosensors, 2021, doi:10.3390/bios11100394_

Round 1
Reviewer 1 Report
The manuscript is well written to express the rationale and quality. It is worth to publish this manuscript in the biosensors. However, have some comments for this manuscript listed as following:
1. The first time to sue an abbreviation in the text, please present both the spelled-out version and the short form:
#Abstract (line 29): EVs
# Line 133:ctDNA
Please carefully check it throughout manuscript.
2. Line 192 ” such as mRNA, microRNA, cfDNA and…” and in the subtitle “3.1. Circulating Cell-Free DNA (CfDNA)” (line 193). The use of abbreviation for cfDNA is inconsistent. Please correct it.
3. Also, in line 191-192, the author mentioned “…which contains CTCs and circulation nucleic acids…, cfDNA and extracellular vehicles (EVS). (EVS should be EVs) The authors seem to make three main aspects three independent paragraphs 3.1-3.3 to illustrate this topic. However, the title of 3.3 subsection is “Exosomes”. It should be used for consistency. Moreover, extracellular vesicles (EVs) included exosomes and microvesicles. Exosomes is the smallest EVs. It will be beneficial to the readers if the authors provide the precise illustration in the difference and correlation between EVs and exosomes.
4. In 3. Liquid biopsy as a new technique section, the “3.3 exosomes” is too brief and there is just one literature citation in the present section. Several reports support the pivotal role of exosomes in tumorigenesis, mediating the immune response, and MRD monitoring. The authors should discuss about the importance of the use of EVs in order to better understand the biology and malignant process of NSCLC. The reference entitled: “Serum exosome microRNA as a minimally invasive early biomarker of AML” doi: 10.1038/srep11295, “Exosomal proteins as diagnostic biomarkers in lung cancer.” doi: 10.1016/j.jtho.2016.05.034, “miR-146a inhibits cell growth, cell migration and induces apoptosis in non-small cell lung cancer cells” doi: 10.1371/journal.pone.0060317, and “MiR-146a-5p level in serum exosomes predicts therapeutic effect of cisplatin in non-small cell lung cancer” should be added to the manuscript.
Author Response
We appreciate the reviewer's help and comments for improving our manuscript. Please see my attached word file "Reviewer1comments_v2.docx"

Reviewer 2 Report
Manuscript ID: biosensors-1402499
Title: Liquid biopsy-based biosensors for MRD detection and treatment monitoring in Non-small cell lung cancer (NSCLC)
Authors: Parvaneh Sardarabadi , Amir Asri kojabad , Davod Jafari , Cheng-Hsien Liu *
In this review, the authors described a comprehensive introduction and discussion of the role of liquid biopsy for MRD detection in NSCLC. This review contains massive information; however, it was quite difficult to find the point of contact between the description of previous research cited in this manuscript and the author’s perspectives. The reviewer thinks that the authors need to clearly state the topic of this manuscript what the authors want to propose. In addition, most importantly, some inaccurate explanations of previous research cited in this manuscript were found, which leads to being lacking in the reliability of the manuscript. Taken together, it was very difficult to find the necessity of the paper to be published in Biosensors. The reviewer would like to encourage resubmissions following extensive revisions.
Major comment:
- There is insufficient analysis and contextualization of the various technologies. The reviewer would like to see more of the author’s perspective on the cited technologies. Currently, many intriguing technologies are listed, but the reviewer is not given much help from the authors to understand the author’s point of view. The reviewer thinks that the authors lapsed into a result reporting mode; please describe the pros and cons of reported technologies, especially for MRD detection in NSCLC as the title of the manuscript is.
- It is pretty hard to understand that the listed technologies are specifically useful for MRD detection, except generally for cancer-clinical applications. Could the authors explain how each listed technology and/or study can specifically apply to MRD detection in NSCLC?
- The part of “Future and prospective of liquid MRD” is not organized and not described well. There are general issues with the language and writing. These do impair the overall understanding of this review’s general point of view.
- The reviewer found that some description of the previous research cited in this manuscript was scientifically wrong. The authors should check all the content, result and meaning, conclusion of the referred researches carefully. For example, in Table 1, the authors described that “CD63” is a biomarker for lung cancer monitoring by citing the reference [56]. However, “CD63”-aptamer was used as a recognition element to capture the A549 exosomes since CD63 is a representative exosome marker.
Minor comment:
- The authors used a capital letter mixed with small letters incorrectly. It may make it difficult for a readership. For example, even in the title of the manuscript, the authors put the capital letter “N” in-between small letters (“…treatment monitoring in Non-small cell lung cancer (NSCLC)”). Please keep the uniformity.
- The authors described “exosome” mixed with “extracellular vesicles”. Exosome may not be equal to extracellular vesicle in the liquid biopsy field.
- In figure 2, the authors schematized “single cell analysis” in the “CTCs” part. Could the authors specify what “single cell analysis” is? That is not the same level of the category with RNA expression, protein biomarkers expression, etc. In addition, in the “cfDNAs” part, the authors listed “1. Exesomes and proteins, 2. ctRNAs, 3. ctDNAs”. First of all, there is a typo that “Exesomes” should be “Exosomes”. Most importantly, these two listed markers (“1. Exosomes and proteins, 2. ctRNAs”) should not be sub-classified under “cfDNAs”. cfDNAs is DNA, neither “Exosomes”, “proteins”, “ctRNAs”.
- In the part of "4. Fluorescence-based biosensing", from line 465 to 471, the authors suddenly described what the exosome research is; the disadvantage of ultracentrifugation, and the requirements for some exosome research, even though this part should focus on the fluorescence-based biosensing as the title is.
Author Response
We appreciate the reviewer's help and comments for improving our manuscript. Please see my attached word file "Reviewer2_comments_v2.docx"

Round 2
Reviewer 1 Report
The revision could be accept in the current form.
Author Response
Thank reviewer!
Reviewer 2 Report
The reviewer appreciates the revisions following the comments. However, there are still some minor mistakes in the revised manuscript. Please check it again and correct several mistakes before publication.
Author Response
We did carefully proofread and revise the manuscript again. We hope we found the "minor mistakes" which the reviewer found. In the resubmitted revised manuscript, we highlight the changes/ and corrections in yellow marks.